# A Nationwide Study of Residual Fate of Fluxapyroxad and Its Metabolites in Peanut Crops Across China: Assessment of Human Exposure Potential

**DOI:** 10.3390/molecules28010194

**Published:** 2022-12-26

**Authors:** Xi Wang, Li Chen, Xin Ren, Shanshan Kang, Wei Li, Zenglong Chen

**Affiliations:** 1State Key Laboratory of Integrated Management of Pest Insects and Rodents, Institute of Zoology, Chinese Academy of Sciences, Beijing 100101, China; 2School of Life Sciences, Hebei University, Baoding 071002, China; 3Beijing Advanced Innovation Centre for Food Nutrition and Human Health, Beijing Technology and Business University, Beijing 100048, China

**Keywords:** fluxapyroxad, nationwide trials, UHPLC–MS/MS, environmental fate, dietary risk

## Abstract

Elaborating on the residual fate of fluxapyroxad and its metabolites based on their nationwide application was vital to protect the human population from their hazardous effects. In this study, a rapid and sensitive analytical method was developed to trace fluxapyroxad and two of its metabolites in peanut matrices using an ultrahigh chromatography method coupled with mass spectrometry (UHPLC–MS/MS) within 3.5 min. The occurrence, pharmacokinetic degradation and terminal magnitudes of fluxapyroxad were reflected in the original deposition of 8.41–38.15 mg/kg, half–lives of 2.5–8.6 d and final concentrations of 0.004–37.38 mg/kg in peanut straw. The total concentrations of fluxapyroxad in peanut straw (0.04–39.28 mg/kg) were significantly higher than those in peanut kernels (<0.001–0.005 mg/kg) and an obvious concentration effect was observed in fresh (0.01–11.56 mg/kg) compared dried peanut straw (0.04–38.97 mg/kg). Fluxapyroxad was demethylated to 3–(difluoromethyl)–*N*–(3′,4′,5′–trifluoro[1,1′–biphenyl]–2–yl)–1H–pyrazole–4–carboxamide (M700F008, 0.02–5.69 mg/kg) and further *N*–glycosylated to 3–(difluoromethyl)–1–(*ß*–D–glucopyranosyl)–*N*–(3′,4′,5′–triflurobipheny–2–yl)–1H–pyrzaole–4–carboxamide (M700F048, 0.04–39.28 mg/kg).The risk quotients of the total fluxapyroxad for the urban groups were significantly higher than those for the rural groups, and were both negatively correlated with the age of the groups, although both acute (ARfD%, 0.006–0.012%) and chronic (ADI%, 0.415–1.289%) risks are acceptable for the human population. The high-potential health risks of fluxapyroxad should be continuously emphasized for susceptible toddlers (1–3 years), especially those residing in urban areas.

## 1. Introduction

The peanut is among the most widely commercial oil crops in the world with import values of 92 thousand dollars and export values of 15 thousand dollars in 2021 [1]. The products of peanut kernel and straw are rich in carbohydrate and dietary fibres, which have been used for oil extraction, livestock roughage and chemical materials [2,3]. However, the stem base of peanut is vulnerable to sclerotium blight damages during field cultivation with up to 80% yield losses with severe infections [4,5]. Fluxapyroxad, 3–(difluoromethyl)–1–methyl–*N*–(3′,4′,5′–trifluoro[1,1′–biphenyl]–2–yl) pyrazole–4–carboxamide, is a novel succinate dehydrogenase inhibitor (SDHI) fungicide, with a high octanol–water partition coefficient (Kow) of 1.35 × 10^−3^ and a low saturated vapour pressure of 2.7 × 10^−6^ mPa at 20 °C [6]. The extraordinary bactericidal activity of fluxapyroxad was proven and could effectively eliminate the peanut sclerotium blight through stunting spore germination, germ tube and mycelia growth of *Sclerotium rolfsii Sacc*. It has been registered in more than 70 crops in China, the European Union (EU), the United States, Japan and Korea, etc. [7].

Notably, fluxapyroxad was classified as a Ⅲ grade of toxicological concern with unexpected negative effects on the survival of non-targeted organisms and human health [8]. Fluxapyroxad posed a high exposure threat to aquatic organisms with a lethal concentration of 50% (LC_50_) of 0.29–0.546 mg a.s./L and to liver tumours developing in rats at 11–82 mg/kg bw/day [9]. Moreover, two main metabolites of fluxapyroxad were reported, namely 3–(difluoromethyl)–*N*–(3′,4′,5′–trifluoro[1,1′–biphenyl]–2–yl)–1H–pyrazole–4–carboxamide (M700F008) and 3–(difluoromethyl)–1–(*ß*–D–glucopyranosyl)–*N*–(3′,4′,5′–triflurobipheny–2–yl)–1H–pyrzaole–4–carboxamide (M700F048) [10], both of which performed toxicological significance in analytical chemistry and risk assessments [11,12]. Evidence through previous studies had revealed a series of analytical methods for tracing IPU and its metabolites, which was in urgent need in peanut matrices with favourable sensitivity [12,13,14]. Furthermore, a persistent characteristic was observed in soil and benthic sediment with a DT_50_ of 183–1000 d and 420–731 d [9]. The dissipation behaviours were also investigated focusing on the parent compound of fluxapyroxad in apples, with half–lives of 9.4–36.5 d [15]. The pharmacokinetic metabolism of fluxapyroxad was unclear currently, which was an important fate pathway in the agroecosystem. Additionally, the dietary risk was evaluated relying on the exposure levels of the parent compound only, with risk quotients of no more than 7.3% [10,16], which did not meet the risk assessment requirement of fluxapyroxad. The residue of fluxapyroxad was defined as the sum of fluxapyroxad, M700F008 and M700F048, and expressed as the parent equivalents for the estimation of the dietary intake for plant commodities [10,11]. Therefore, it was significant to elaborate upon the environmental fate of fluxapyroxad and its metabolites in peanut crops to ensure the food safety of peanut consumption and to precisely evaluate the dietary risks humans could be exposed to.

The present study aimed to develop a rapid and effective analytical method to trace fluxapyroxad and its metabolites in peanut kernels and straws; highlighting the fate tendency and dietary risks of the total fluxapyroxad based on its nationwide applications in peanut ecosystems. An analytical method was developed for the simultaneous determination of fluxapyroxad, M700F008 and M700F048 in peanut kernels and straw matrices. Moreover, the occurrence, degradation and the metabolism of fluxapyroxad were elucidated upon with regard to the original deposition (C_0_), half-life (T_1/2_) and the concentration variation in large-scale trials across China, for which impossible influencing factors were further discussed. Furthermore, the terminal levels of fluxapyroxad were investigated both in fresh and dried peanut crops in large-scale fields across China. A systematic analysis was conducted between highest residue (HR) of fluxapyroxad and maximum residue limits (MRLs) established internationally providing early warning suggestions for the import and export trade of peanut crops. Finally, the dietary risks for population groups in terms of age, region and gender were evaluated to provide the basic data for scientific use and an effective supervision of fluxapyroxad in peanut crops, providing references for protecting the health of the consumer population from agrochemical exposures.

## 2. Results and Discussion 

### 2.1. Method Validation for Tracing Fluxapyroxad and Metabolites in Peanut Matrices 

The specificity, linearity, matrix effect (ME), limit of detection (LOD), limit of quantitation (LOQ), repeatability, reproducibility and stability were positively validated based on the analytical quality control criteria of SANTE/11312/2021 [17]. The specificity of the developed method was verified by blank assays of peanut kernel and straw samples from control plots with no detectable interference at retention times of fluxapyroxad (2.55 min), M700F008 (2.43 min) and M700F048 (2.11 min), which were requiring 2/3–4/5 of the time when compared to the previous studies [12,13,14,15]. Satisfactory linearities were observed from 0.001 to 0.5 mg/kg by measurement of calibration curves obtained from solvent MeCN and matrix extracts, with reference to correlation coefficients (R^2^) of 0.9967–0.9999 in peanut kernel and 0.9996–0.9999 in peanut straw. The straw samples with high concentrations of the target compounds were diluted uniformly within the linearity range. Significant differences were observed for MEs of fluxapyroxad and metabolites ranging from −30.3% to 10.6% in peanut kernels and ranging from –48.3% to –41.3% in peanut straw, which might be positively related to the organic compounds contained in both the peanut kernels and straw [18], and the concentration of the target compounds in the samples [15,19]. Significantly the matrix suppression effects were observed for fluxapyroxad (−30.3–−48.3%) and M700F008 (−16.2–−47.8%) both in peanut kernels and peanut straw, and for M700F048 (−41.3%) in peanut straw, whereas the matrix enhanced effect for M700F048 was 10.6% in peanut kernels. Consequently, the external matrix–matched standards were selected for the accurate quantitation of the fluxapyroxad, M700F008 and M700F048 to reduce the matrix effect and obtain the true detection data in the sample. The LODs of the fluxapyroxad, M700F008 and M700F048 were 8.0928 × 10^−7^–0.0059 mg/kg, 3.8604 × 10^−8^–0.0194 mg/kg and 2.0358 × 10^−7^–0.0005 mg/kg, respectively, which were determined at a signal-to-noise (peak to peak) ratio of three under the above conditions. The LOQ was estimated to be 0.001 mg/kg in the associated matrixes, which was taken as the lowest concentration level with a satisfactory recovery (70–120%) and relative standard deviations (RSDs) of less than 20% (Table 1), providing a 2–3.4–fold increase in sensitivity [12,13,14,15]. Recovery experiments were performed by spiking a standard working solution of fluxapyroxad, M700F008 and M70F048 at low (LOQ), medium (50–fold LOQ) and high levels (500–fold LOQ and an additional spiked 50 mg/kg for the peanut straw) to cover the maximum concentrations of the samples in six replicates. The mean recoveries and RSDs of ethirimol were estimated to be 79–112% and 5.4–13.0% in the peanut kernels and 86–106% and 2.6–16.0% in the peanut straw (Table 1). The stability of the fluxapyroxad, M700F008 and M70F048 were also evaluated weekly in stock standard and matric–matched solutions during the experiments, and no significant differences were found for the concentrations of fluxapyroxad in the study with p more than 0.05 at a 95% probability. Typical UHPLC–MS/MS (MRM) chromatograms of fluxapyroxad, M700F008 and M700F048 are shown in Figure 1A–C.

### 2.2. Occurrence and Degradation of Fluxapyroxad during Peanut Cultivation

The initial concentrations of fluxapyroxad in peanut straw ranged from 8.41 to 38.15 mg/kg, and those were almost undetected in the peanut kernel (<0.001–0.005 mg/kg) after foliage application. The higher concentration of fluxapyroxad was observed in *Yvhua 22 peanut* straw (C_0_, 38.15 mg/kg), which was 4.5 times over than that of *Guihua 17 peanut* (C_0_, 8.41 mg/kg), demonstrating that peanut cultivar can affect the deposition of fluxapyroxad. The former showed higher straw of 43 cm and full kernel of 81.6 g than the latter cultivar [20]. The initial deposition of M700F048 in the peanut straw (0.09–0.69 mg/kg) was relatively higher than that of M700F008 (0.02–0.29 mg/kg), indicating that fluxapyroxad can be preferentially metabolized to M700F008 in peanut straw, and further liable to *N*–glycosylated to M700F048.

Moreover, the pharmacokinetic degradation of fluxapyroxad conformed to pseudo–first–order kinetics well with the determination coefficient (r) of 0.7819–0.9936. The T_1/2_ of fluxapyroxad was 2.5–8.6 d in the peanut straw, which was shorter than that of the apple with 9.4–12.6 d [15]. The shortest T_1/2_ of fluxapyroxad was observed in Trial #10 (107.45 °E, 22.83 °N), and the longest one was observed in Trial #9 (113.25 °E, 28.27 °N). Therefore, the high temperature and rainfall promoted the pharmacokinetic degradation of fluxapyroxad in peanut crops. The concentration of M700F048 in peanut straw was 0.02–8.62 mg/kg during peanut cultivation, which was up to 14.4 times more than that of M700F008 (0.01–4.12 mg/kg), demonstrating that *N*–glycosylation was the main metabolized process of fluxapyroxad in the peanut straw. The maximal concentration of M700F048 was found in Trial #9 (113.25 °E, 28.27 °N), whereas that of M700F008 was observed in Trial #5 (113.68 °E, 34.96 °N). The concentration variations of the metabolites were consistent with those of the parent.

### 2.3. Terminal Magnitude and MRL Comparison of Fluxapyroxad in Peanut 

The terminal concentrations of fluxapyroxad were 0.004–37.38 mg/kg in the peanut straw, accounting for 0.2–64.4% of the original concentration. The relatively low levels were observed in the peanut kernel with <0.001–0.002 mg/kg. The study for fluxapyroxad in wheat cultivation systems further confirmed the result, where the concentration of fluxapyroxad in wheat straw (0.31 mg/kg) was significantly higher than that in wheat grain (<0.01 mg/kg) [10]. The concentrations of fluxapyroxad in fresh peanut straw (0.004–11.10 mg/kg) were far lower than that in dried peanut straw (0.004–37.38 mg/kg), with amazing water contents of 67–79% in the fresh samples. The concentration of M700F008 was 0.006–1.94 in fresh peanut straw and 0.02–4.11 mg/kg in dried peanut straw, while they were 0.006–1.94 and 0.03–8.62 mg/kg for M700F048, respectively. There was a necessity to pay more attention to the harm of the concentrated fluxapyroxad in the dried peanut straw to livestock or in the agricultural production process. Moreover, the final concentrations of M700F048 (0.03–8.62 mg/kg) were higher in comparison with that of M700F008 (0.02–4.11 mg/kg) in the peanut straw, indicating that the *N*–glycosylation was the main metabolized process in the peanut straw. Typical UHPLC–MS/MS (MRM) chromatograms of fluxapyroxad, M700F008 and M700F048 are shown in Figure 1D–F.

The total concentrations of fluxapyroxad, M700F008 and M700F048 were further investigated by factors of 0.72–1.039, ranging from <0.001 to 0.005 mg/kg in the peanut kernel and between 0.04–39.28 mg/kg in the peanut straw (Table 2). The STMRs of the total fluxapyroxad were 0.003 mg/kg and 1.28 mg/kg in both the peanut kernel and peanut straw, corresponding to the HRs, which were 0.005 and 39.28 mg/kg, respectively. Additionally, the terminal magnitudes of the total fluxapyroxad were discussed with the influencing factors of climate conditions, soil characteristics and peanut cultivars in this study. High temperature (29.3 °C) and precipitation (1430 mm) in Trial #10 promoted the degradation of fluxapyroxad (Figure 2A,B). The relatively low deposition of the total fluxapyroxad in peanut crops was observed in Trial #10 (0.04–0.35 mg/kg) in comparison with that in Trial #7 (3.64–39.28 mg/kg) and Trial #9 (10.22–20.84 mg/kg). This was mainly due to the high temperature, which accelerated the evaporation, codistillation and thermodegradation [21], and the high rainfall, which increased the amounts of leaching and runoff of the agrichemicals in the soil [22]. Moreover, the higher organic matter (OM%) in Trial #1 (3.8%) was beneficial for the accumulation of fluxapyroxad in the peanut crops with a concentration of 1.43–1.74 mg/kg (Figure 2C), which was higher than that in Trial #2 and Trial #10 with 0.04–0.68 mg/kg. This result was because of the strong adsorption of hydrophobic fluxapyroxad in the soil with high organic matter [23]. Furthermore, the alkaline soil conditions promote the degradation of fluxapyroxad based on the weak acidity of fluxapyroxad (pH, 5–6) (Figure 2D). The concentration of the total fluxapyroxad was 0.95–1.28 mg/kg in Trial #3 with the peak pH value of soil (8.6), which was lower than that in Trial # 1 and Trial #9 (1.43–18.01 mg/kg) with pH values of 5.4–7.9. Moreover, the maximum concentration of fluxapyroxad in peanut crops were observed in *Luhua 14 peanut* of 3.64–39.28 mg/kg, whereas the minimum was in *Guihua 17 peanut* with 0.04–0.35 mg/kg, and the terminal concentrations of fluxapyroxad in *Baisha 1016 peanut* were significantly different in Trial #9 (10.22–18.01 mg/kg) and Trial #8 (1.26–1.85 mg/kg). The specific mechanism and possible factors synergistically influencing the residual fate of fluxapyroxad require further investigation in follow–up studies.

The MRL values of fluxapyroxad was 0.01 mg/kg in the peanut kernels in China, which were consistent with the values of Codex Alimentarius Commission (CAC) [24], America [25], Japan [26] and so on. The residual concentrations of fluxapyroxad were lower than the MRLs in the peanut kernel, which would not affect the import and export trade of peanut kernels. However, there was no corresponding values of fluxapyroxad in peanut straw with extremely high terminal levels, which were directly used as livestock and poultry roughage and having a potential high risk of exposure to animals and even human health. Therefore, it is suggested to speed up formulating a MRL of fluxapyroxad in peanut straw to avoid the exposure risks.

### 2.4. Dietary Risks of Total Fluxapyroxad under Large–Scale Applications 

To assess acute dietary exposure of the total fluxapyroxad through peanut kernel consumption, the ARfD% of the total fluxapyroxad was estimated for different populations, including children (1–11 years old), general population (>11 years old) and childbearing women (14–45 years old) (Figure 3A). The acute dietary risks of the total fluxapyroxad in peanut kernels was 0.006–0.012% calculated by the deterministic model. The maximum acute dietary risk of fluxapyroxad was also observed in children with 0.012%, followed by 0.006% for both general population and childbearing women, which was closely related to the high estimated short-term intake for children and should be brought to the forefront of the public attention. These results preliminarily show that there was a risk dilution during the combined exposure to fluxapyroxad in multiple crops and a low possibility was observed in consumers of dietary peanut kernels.

The chronic dietary risk quotient of fluxapyroxad was also calculated in registered crops with 0.415–1.350%. The minimal ADI% of the total fluxapyroxad was observed in peanut kernels with 0.007–0.023% for the association population (Figure 3B), accounting for 1.7% of the total dietary risk values of fluxapyroxad in registered crops. Moreover, the ADI% of fluxapyroxad was increased in banana, cucumber and brown rice, which were 0.044–0.156%, 0.109–0.409% and 0.218–0.780%, respectively. This case demonstrated that the dietary risks of the total fluxapyroxad could be accumulated in multiple crops. The total chronic dietary risks were further analysed in various consumers based on the high acute dietary risks for children, including different age groups (toddlers of 2–3 years, children of 4–10 years, teenagers of 11–17 years and adults of ≥18 years), region groups (urban and rural) and sex groups (male and female) (Figure 3C). First, the dietary risks of fluxapyroxad were negatively correlated with age in the life cycle. The ADI% reached the highest value in toddlers with 1.350%, and the minimum was observed in seniors with 0.415%, possibly due to the highest NEDI in toddlers of 0.26 × 10^−3^ mg/kg bw [27], with toddlers being more vulnerable to daily health exposure [28,29]. Moreover, the ADI% of the urban consumers (0.504–1.350%) was higher than those of the rural groups (0.415–0.936%) with *p* < 0.01 at a 95% probability, which was consistent with the exposure of ethirimol in peanuts [2]. Additionally, there was no difference in the dietary risks between males and females, which ranged from 0.450% to 1.289% for males and from 0.415% to 1.350 % for females. The maximum and minimum dietary risks for males and female were reached in the same groups, which were in urban toddlers and rural seniors, respectively, resulting from the equal dietary structure between males and females.

## 3. Materials and Methods

### 3.1. Chemicals, Reagents and Standard Solutions

Fluxapyroxad standard (purity 99.99%), M700F008 standard (purity 99.5%), and M700F048 standard (purity 94%) were purchased from Dr. Ehrenstorfer (Augsburg, Germany). The formulation product 167 g/L of the fluxapyroxad suspension concentrate (SC) was provided by J&K Scientific (Beijing, China). Formic acid (FA) of chromatographic grade and acetonitrile (MeCN, ≥98.0% purity) were obtained from Sigma–Aldrich (Steinheim, Germany). Analytical grade MeCN (99.8% purity), acetic acid (AcOH), sodium chloride (NaCl) and anhydrous magnesium sulfate (MgSO_4_) were purchased from Beijing Chemical Company (Beijing, China). Ultrapure water was prepared from a Milli–Q system (Bedford, MA, USA). Octadecylsilane (C18, 40–60 µg) and graphitized carbon black (GCB, 120–400 Mesh) were provided by Agela Technologies (Tianjin, China). 

The working solution and calibration were prepared using the appropriate dilution of the stock solution on the day of analysis, and the mixed-matched standard solution of fluxapyroxad, M700F008 and M700F048 with concentrations of 0.001, 0.005, 0.01, 0.025, 0.05, 0.1, 0.25 and 0.50 mg/L were prepared by adding blank sample extracts to each serially diluted standard solution. The stock and working solutions were stored at –20 and 4 °C under dark conditions, respectively. 

### 3.2. Plant Care and Pesticide Application

Ten cultivation origins of peanut crops were selected from the main producing provinces or municipalities of China spanning five climate zones, which had no history of fluxapyroxad in the past three years. Based on the Organization for Economic Co-operation and Development Guidelines (OECD) for the Testing of chemicals No. 509 Crop Field Trial [30], three trial plots were chosen to avoid random errors, and one plot was used as a control. Each plot was 50 m^2^ and separated using a buffer area of 0.5 m from the next plot. The commercial formulation of fluxapyroxad was applied at a dosage of 100.2 g of the active ingredient per ha (g a. i./ha) for the peanut crops. The application frequency and interval of pesticides were 2 and 7 d, respectively, according to the good agricultural practices (GAPs) recommendations. The pharmacokinetic dissipation study was conducted in Trial #2, Trial #5, Trial #9 and Trial #10, which have typical peanut varieties and distinct climate conditions across China. The geographic information, hydrological characteristics, climate factors, soil characteristic and radish cultivars are summarized in Table 3.

### 3.3. Sample Collection and Preparation

Representative peanut samples were obtained from each plot, excluding the edges and ends of the plots, preferably with intact rinds and disease-free. Both the peanut kernel and straw samples were collected at increasing intervals on 2 h, 7 d, 14 d, 21 d and 28 d after application for the dissipation investigation. For the terminal level study of fluxapyroxad, the samples of mature peanut grains and straws were collected at preharvest intervals (PHIs) of 14 and 21 d after the last application in accordance with FAO/WHO recommendations. Each of the kernel and straw samples were not less than 1 kg and were shipped to the laboratory within 8 h. Exterior impurities were removed, chopped into quarters within 1 cm, divided into 150 g subsamples and kept frozen at −20 °C.

Representative homogenized samples (5 ± 0.01g) of peanut kernel and straw were weighed into 50 mL polytetrafluoroethylene centrifuge tubes. Five mL of 1.2% AcOH–MeCN and equivalent volumes of ultrapure water were added to the peanut kernel and additional 5 mL solutions were required for the peanut straw. Then the mixture was shaken for 5 min using a CK–2000 high-throughput grinder (TH Morgan, Beijing, China) at an oscillation frequency of 1350 min^−1^. Subsequently, 3 g of NaCl were added and the mixture was repeatedly vortexed for 5 min, followed by centrifugation for 5 min with a Frontier FC5706 centrifuge (OHAUS, Nänikon, Switzerland) at 1107× *g* relative centrifugal force (RCF). Additionally, 1.5 mL of the supernatant was transferred into a single-use centrifuge tube, which contained 50 mg C18 and 150 mg anhydrous MgSO_4_ for the peanut kernel and 50 mg C18, 10mg GCB and 150 mg anhydrous MgSO_4_ for the peanut straw. The vortex step was repeated for 30 s and centrifuged for 1 min with a Heraeus Pico 17 Centrifuge (Thermo Scientific, Am Kalkberg, Germany) at 2188× *g* relative centrifugal force. The resulting supernatant was withdrawn, reconstituted by the initial mobile phase, and filtered through a 0.22 μm nylon syringe filter into an autosampler vial for UHPLC–MS/MS injection.

### 3.4. Instrumentations

An Ultra–performance Liquid Chromatography–Triple Quadrupole Tandem Mass Spectrometry (Waters ACQUITY UPLC H–Class/Xevo TQD) equipped with a Waters ACQUITY HSS T3 column (100 mm × 2.1 mm, 1.8 μm) was employed to separate the fluxapyroxad and main metabolites simultaneously. Gradient elution was initialized by 10/90 volume of MeCN (solvent A) and 0.05% FA in water (solvent B) for 0.5 min, and then increased from 10% to 90% A over 2.0 min, held at 90% A for 0.1 min, and then equilibrated by 10% A for 0.9 min, providing a total analysis time of 3.5 min. The flow rate, column temperature and the injection volume were set at 0.45 mL/min, 40 °C, and 5 µL, respectively. A high sensitivity for fluxapyroxad was optimized under a negative mode with a 1.5 kV capillary voltage, 150 °C source temperature and 350 °C desolvation temperature. The nebulizer gas was N_2_, and the collision gas was Ar with 2 × 10^−3^ mbar in the T-wave cell. Cone and desolvation flows of 50 and 650 L/h were applied, respectively. The multiple reactions monitoring (MRM) transitions m/z 381.3→131.1 and m/z 381.3→248.1 were used for the quantification and confirmation of fluxapyroxad with the collision energy of 26 and 22 eV, whereas they were m/z 366.0→306.0 (20 eV) and 366.0→326.0 (16 eV); and m/z 528.1→326.1 (28 eV) and m/z 528.1→346.0 (20 eV) for M700F008 and M700F048, respectively. The cone voltage of the fluxapyroxad, M700F008 and M700F048 was set at 50, 30 and 56 V, respectively, and the dwell times were 0.034 s. 

### 3.5. Mathematical Calculations

The matrix effect (ME) refers to the variation in ionization efficiency of the target compounds with the presence of co-effluents, which were evaluated using Equation (1):(1)$$\mathrm{ME}=\left(\frac{S_{m}}{S_{S}}-1\right)\times 100\%$$
where *S_m_* and *S_s_* are the slopes of the calibration curves obtained in the matrix and in the solvent, respectively. ME ≤ −10% (MeCN as the reference) indicates a significant matrix suppression effect, whereas ME ≥ 10% indicates a significant matrix enhancement effect. ME is not significant with −10% < ME < 10% and does not exist at a value of zero [31]. The dissipation curves of fluxapyroxad in peanuts were plotted according to the data of the pesticide concentrations; the possible first–order kinetic model can give a good description of the dissipation behaviour and the half–life of degradation (T_1/2_) can be calculated (2, 3). The possible first-order kinetic model is as follows [32]: (2)$$C_{t}=C_{0}\times \exp^{-\mathrm{kt}}\;$$
where C_0_ and C_t_ represent the initial sample concentration (mg/kg) and residue concentration (mg/kg) at time t (d). The T_1/2_ was calculated using Hoskins’ Formula (3):(3)$$T_{1/2}=\frac{\ln2}{k}=\frac{0.693}{k}$$

The starting point of the regressive function is the maximum value of the concentrations and decreases in the following days. The deterministic model is a popular approach for a dietary risk assessment. It is used widely because it requires minimal resources and data to calculate, and it is fairly easy to understand. Additionally, the deterministic model of acute dietary risk is based on the Food and Agriculture Organization of the United Nations (FAO) classification guidelines, there were four different models for the calculation of the national estimated short–term intake of pesticides (NESTI) [33], Formula (4) is applicable to the peanut samples in this study:(4)$$\mathrm{NESTI}=\frac{\mathrm{LP}\times \mathrm{STMR}(\mathrm{STMR}-P)}{bw}$$
(5)$$\mathrm{ARfD}\%=\frac{\mathrm{NESTI}}{\mathrm{ARfD}}\times 100\%$$
where NESTI (mg/kg bw/d) is the estimated short-term intake of the country, STMR–P was the supervised trials median residue of the total fluxapyroxad (sum of fluxapyroxad, M700F008 and M700F048) corrected with a processing factor (Peanut oil). The amounts of M700F008 and M700F048 were converted to fluxapyroxad by factors of 1.039 and 0.72, respectively, which were calculated using the molecular weight of fluxapyroxad (m.w., 381.3) divided by each metabolite M700F008 (m.w., 366.0) and M70F048 (m.w., 528.1). LP (g/d) is the daily consumption of food for 97.5 percent of consumers, and bw is body weight. ARfD (mg/kg bw/d) is the acute reference dose, which for fluxapyroxad is 0.25 mg/kg bw d. ARfD% represents the percentage of the acute reference dose. The deterministic model of chronic dietary risk includes Formulas (6) and (7): (6)$$\mathrm{NEDI}=\frac{\mathrm{STMR}(\mathrm{STMR}-P_{i})\times F_{i}}{bw}$$
(7)$$\mathrm{ADI}\%=\frac{\mathrm{NEDI}}{\mathrm{ADI}}\times 100\%$$
where NEDI is the national estimated exposure intake (mg/kg bw). Fi is the average daily intake of a certain food in China (kg). ADI% is the risk quotient of chronic dietary risk. The acceptable daily intake (ADI) of fluxapyroxad is 0.02 mg/kg bw. The risk is estimated to be acceptable with an ADI% < 100%; however, the risk will bring a long–term health threat with ADI% > 100%. 

### 3.6. Statistical Analysis

The differences in data sets that were normally distributed were assessed by the *t*-test and the one-way Analysis of Variance (ANOVA) with Duncan’s post-hoc test using SPSS 22.0 (IBM corporation, IL, USA). The different tests for nonnormally distributed datasets were accessed by the Kruskal–Wallis one–way ANOVA test. Differences at *p* < 0.05 were considered statistically significant for both Tukey’s HSD test and Student’s *t*-test in this study.

## 4. Conclusions

A rapid and sensitive tracing methodology was developed for fluxapyroxad, M700F008 and M700F048 in peanut matrices by UHPLC–MS/MS, with an LOQ of 0.001 mg/kg and retention times of 2.11–2.55 min. The occurrence, dissipation and concentration variations of fluxapyroxad were further elucidated in peanut crops, as reflected by the original deposition, half-lives and terminal levels. Fluxapyroxad was metabolized to M700F008 in peanut crops, which can be the precursor of *N*–glycosylated M700F048. The higher concentrations of the total fluxapyroxad were observed in the peanut straw as opposed to the kernel, especially in dried straw with concentrations of up to 38.97 mg/kg. The acute risks of the total fluxapyroxad were of 0.006–0.012%, whereas the chronic risks were from 0.415% to 1.289% for human population. Despite the acceptable exposure risks, the potential health hazards of fluxapyroxad should be continuously emphasized, given its increasing applications.

## Figures and Tables

**Figure 1 molecules-28-00194-f001:**
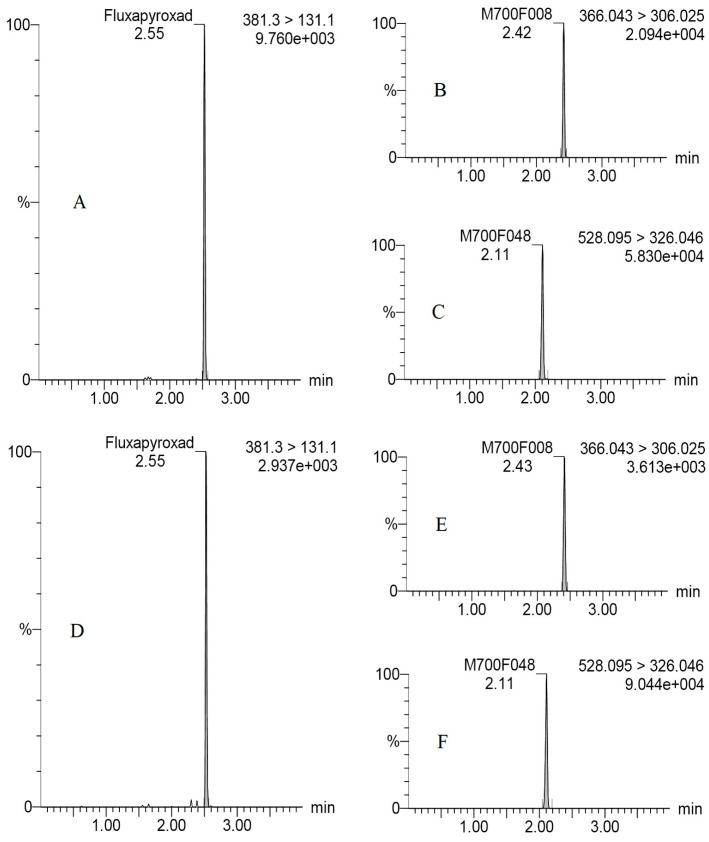
Typical UHPLC–MS/MS (MRM) chromatograms of fluxapyroxad, M700F008, and M700F048 in 0.05 mg/L of matric–matched standard solution (**A**–**C**) and peanut straw at 21 days in Trial #3 (**D**–**F**).

**Figure 2 molecules-28-00194-f002:**
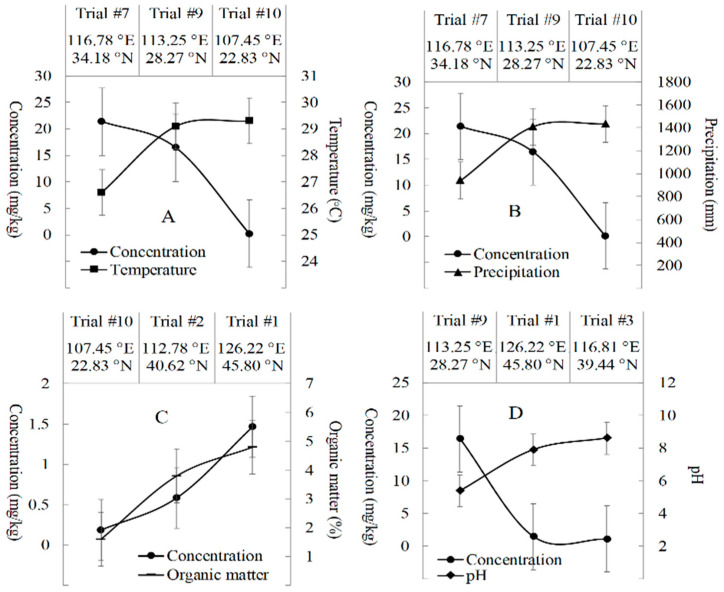
Effects of environmental conditions for terminal concentrations of total fluxapyroxad in different trial sites of peanut crops. (**A**) Temperature was negatively correlated with concentration of fluxapyroxad; (**B**) Precipitation was negatively correlated with concentration of fluxapyroxad; (**C**) Organic matter content was positively correlated with concentration of fluxapyroxad; (**D**) pH was negatively correlated with concentration of fluxapyroxad.

**Figure 3 molecules-28-00194-f003:**
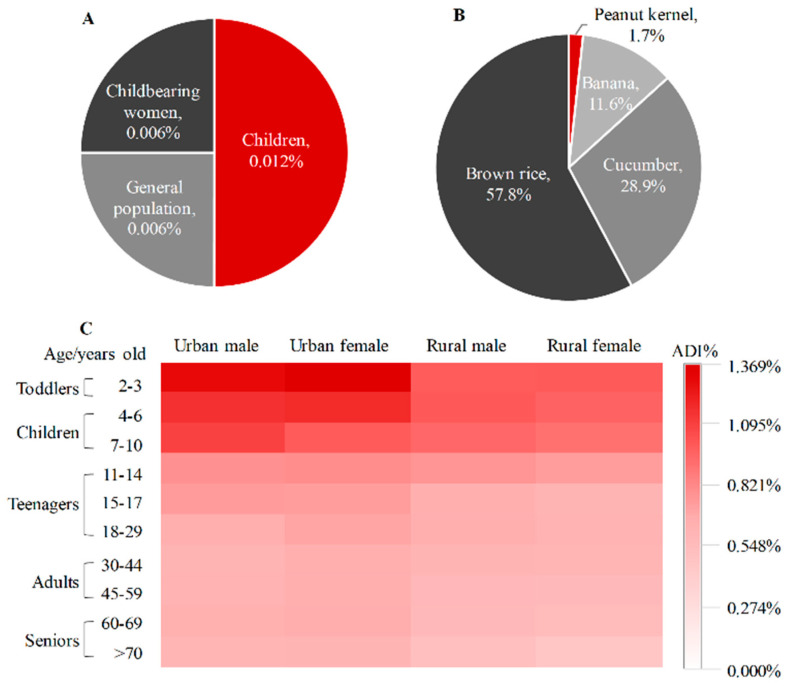
(**A**) Acute dietary risks of total fluxapyroxad in peanut kernel for three group consumers; (**B**) proportion of exposure risk of total fluxapyroxad in registered crops; (**C**) chronic dietary risks of total fluxapyroxad in crops for various groups consumers.

**Table 1 molecules-28-00194-t001:** Linear regression parameters and recoveries for the fluxapyroxad, M700F008 and M70F048 in different matrices.

Compound	Matrix	Regression Equation	R^2^	Matrix Effect (%)	LOQ (μg/kg)	Mean Recoveries ± SD (%, *n* = 6)		
						Low	Medium	High
**Fluxapyroxad**	Acetonitrile	y = 1309x + 27.166	0.9991	–	–	–	–	–
	Peanut kernel	y = 4400.2x + 34.488	0.9999	−30.3	0.001	110 ± 6.9	99 ± 13.0	98 ± 6.3
	Peanut straw	y = 2532.1x + 46.288	0.9996	−48.3	0.001	105 ± 16	86 ± 8.4	105 ± 3.1
**M700F008**	Acetonitrile	y = 30,737x + 652.69	0.9994	–	–	–	–	–
	Peanut kernel	y = 74,053x + 751.33	0.9999	−16.2	0.001	95 ± 5.4	112 ± 8.7	96 ± 11.0
	Peanut straw	y = 58,882x + 745.08	0.9999	−47.8	0.001	102 ± 3.1	103 ± 3.3	106 ± 2.6
**M700F048**	Acetonitrile	y = 15,270x + 132.41	0.9999	–	–	–	–	–
	Peanut kernel	y = 33,924x − 374.84	0.9967	+10.6	0.001	79 ± 7.7	99 ± 10.0	85 ± 6.2
	Peanut straw	y = 26,018x + 47.223	0.9996	−41.3	0.001	106 ± 15.0	101 ± 7.1	105 ± 3.1

**Table 2 molecules-28-00194-t002:** Concentrations of fluxapyroxad and metabolites in peanut samples at 14–21 d after application.

**Location**	**Trial Sites**	**Peanut Cultivars**	**Fluxapyroxad**		**M700F008**		**M700F048**		**Fluxapyroxad (Sum)**	
			**Peanut Kernel**	**Peanut Straw**	**Peanut Kernel**	**Peanut Straw**	**Peanut Kernel**	**Peanut Straw**	**Peanut Kernel**	**Peanut Straw**
**Heilongjiang**	Trial #1 (126.22 °E, 45.80 °N)	*X* *iao* *lihong*	≤0.002	0.49–0.60	0.002–0.003	0.22–0.32	≤0.001	0.54–1.17	0.004–0.005	1.43–1.74
**Inner Mongolia**	Trial #2 (112.78 °E, 40.62 °N)	*Luhua No.11*	≤0.001	0.22–0.47	<0.001	0.05–0.06	<0.001	0.18–0.29	0.003	0.40–0.68
**Beijing**	Trial #3 (116.37 °E, 39.44 °N)	*Ji You 4*	<0.001	0.12–0.22	<0.001	0.16–0.29	<0.001	0.81–1.09	0.003	0.95–1.28
**Shandong**	Trial #4 (117.12 °E, 36.49 °N)	*Luhua No.14*	≤0.001	0.05–0.23	≤0.001	0.09–0.15	<0.001	0.50–0.92	0.003	0.50–1.05
**Henan**	Trial #5 (113.68 °E, 34.96 °N)	*Yuhua 22*	≤0.001	6.51–29.67	<0.001	1.69–5.69	<0.001	1.29–3.13	0.003	10.51–33.04
**Sichuan**	Trial #6 (103.10 °E, 30.95 °N)	*Tianfu 9*	0.002–0.003	0.43–0.87	0.001	0.17–0.33	<0.001	0.29–0.81	0.003–0.005	0.82–1.80
**Anhui**	Trial #7 (116.78 °E, 34.18 °N)	*Luhua No.14*	≤0.001	1.40–37.38	≤0.001	0.42–0.77	≤0.001	1.53–2.60	0.003	3.64–39.28
**Hubei**	Trial #8 (114.32 °E, 30.48 °N)	*Baisha 1016*	≤0.001	0.61–0.84	≤0.001	0.34–0.54	<0.001	0.42–0.83	0.003	1.26–1.85
**Hunan**	Trial #9 (113.25 °E, 28.27 °N)	*Baisha 1016*	<0.001	5.53–12.31	≤0.001	0.91–1.82	≤0.001	5.21–9.23	0.003	10.22–20.84
**Guangxi**	Trial #10 (107.45 °E, 22.83 °N)	*Guihua 17*	≤0.001	0.004–0.07	0.001–0.002	0.02–0.05	≤0.001	0.03–0.32	0.003–0.004	0.04–0.35

**Table 3 molecules-28-00194-t003:** Crop cultivars, climate characteristics and soil physicochemical properties of trial sites in the study.

Trial Sites	Crop Cultivars	Climate Factors			Soil Properties		
		Climate Type	A.T. ^a^ (°C)	A.R. ^b^ (mm)	Soil Type	pH	O.M. ^c^ (%)
Trial #1 *Zhaodong, Heilongjiang* (126.22 °E, 45.80 °N)	*Xiaolihong*	Cold temperate monsoon climate	20.8	524	Phaeozem	7.9	4.8
Trial #2 *Wulanchabu, Inner Mongolia* (112.78 °E, 40.62 °N)	*Luhua 11*	Mid–temperate continental monsoon climate	18.4	300	Clay	8.2	3.8
Trial #3 *Tongzhou, Beijing* (116.81 °E, 39.44 °N)	*Jiyou 4*	Warm temperate subhumid continental monsoon climate	25.2	650	Loam	8.6	1.6
Trial #4 *Jinan, Shandong* (117.20 °E, 36.49 °N)	*Luhua 14*	Warm temperate continental monsoon climate	26.4	628	Loam	7.1	1.0
Trial #5 *Xinxiang, Henan* (113.68 °E, 34.96 °N)	*Yuhua 22*	Warm temperate continental monsoon climate	25.6	606	Chao soil	7.1	1.2
Trial #6 *Pengzhou, Sichuan* (103.10 °E, 30.95 °N)	*Tianfu 9*	Subtropical humid climate	25.1	1003	Loam	6.5	1.5
Trial #7 *Suzhou, Anhui* (116.78 °E, 34.18 °N)	*Luhua 14*	Warm temperate subhumid monsoon climate	26.6	941	Sandy loam	7.4	1.1
Trial #8 *Wuhan, Hubei* (114.32 °E, 30.48 °N)	*Baisha 1016*	Subtropical monsoon climate	28.8	1050	Sandy loam	6.8	1.9
Trial #9 *Changsha, Hunan* (113.25 °E, 28.27 °N)	*Baisha 1016*	Subtropical monsoon climate	29.1	1410	Clay	5.4	2.7
Trial #10 *Nanning, Guangxi* (107.45 °E, 22.83 °N)	*Guihua 17*	Subtropical monsoon climate	29.3	1435	Laterite	6.4	1.6

^a^ Annual Temperature, ^b^ Annual Rainfall, ^c^ Organic Matter.

## Data Availability

The data presented in this study are available on request from the corresponding authors.

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
