# Peer review of "A Nationwide Study of Residual Fate of Fluxapyroxad and Its Metabolites in Peanut Crops Across China: Assessment of Human Exposure Potential"

_molecules, 2022, doi:10.3390/molecules28010194_

Round 1

Reviewer 1 Report

This study investigates the residual occurrence of the fluxapyroxad and its metabolites in peanut crops in China. A UHPLC-MS/MS method was devised to rapidly and simultaneously determine the fluxapyroxad and two metabolites, M700F008 and M700F048,  in peanut kernel and straw matrices. It is reported that during method validation, no interference at target compound retention times was detected, calibration curve linearity was observed in the acceptable concentration range, and significant ionization suppression was detected for fluxapyroxad and M700F008. At the same time, the plant matrix had an enhancing effect on the ionization of  M700F048. LOD value is determined at an S/N ratio of 3 and more, while LOQ was selected as the lowest concentration with a recovery level between 70-120%. The LC-MS analysis of samples collected during cultivation at trial sites in China provided information on fluxapyroxad and its two metabolites' occurrence in peanut kernel and straw material. The known metabolic degradation of the fluxapyroxad in plants is the demethylation of the pyrazole moiety. The resulting compound is a precursor for metabolic N-glucosidation.

The Authors should present the metabolic pathway scheme of the fluxapyroxad in plants and comment on the other metabolic products that might be expected. Elaborate more closely why particular attention was put to the risk assessments of these two metabolites exclusively.

In the LC-MS analytical method validation section, it was mentioned that for accurate quantification, external matrix-matched standards were selected. The authors should provide more details about employed standards and calibration procedures in this study.

For reducing the observed matrix effect on the ionization of the compounds, did the authors consider any sample clean-up procedures? Are such clean-up procedures necessary?   

The authors decided to optimize sensitive mass measurement under positive ionization mode. However, it is reported that fluxapyroxad and its metabolites display abundant negative molecular ions with higher responses than in positive ion mode. Is there any particular reason to optimize mass measurement in positive ion mode in this study?  

Author Response

The authors thank the reviewer for spending time to review our manuscript. The constructive comments helped us to improve our work further. The authors have carefully addressed each comment and question you raised in detail, which we hope will meet with your approval. Please refer to the detailed responses as follow.

Reviewer 2 Report

This is a research paper presenting developing a rapid method for analysis of  fluxapyroxad and its two metabolites in peanut and assesing diatery risk.  The research itself is original and interesting, and the manuscript is  well organized and nicely written. On the other hand, the below comments should be taken into account to put across the paper.

Line 72. The ami of the paper is to develop a method to analyse fluxapyroxad and its metabolites in peanut. Therefore, the method used in the literature should have been citied and the lack on this subject should be explained.

Line 134. The amount tof laboratory sample should be specified. The shipping and storage conditions from farm to laboratory should be explained.

Line 177. The reference for evaluation should be provided

Line 169. A chromatogram of a standard and sample should be provided

Line 184. The reference for equation should be provided

Line 220. Experimental design for the method validation study should be given in more detail (number of the replicates for each parameter, spike concentrations, details on repatablity and reproducibility conditions etc)

Line 221. Could you please give full name of an abbrevation at the first place in the manuscript. (ME, LOD, LOQ)

Line 239. The unit for LOD of the method should be mg/kg rather than mg/L

Line 226, 280. The linearity of the method was indicated that it ranged between 0.001 and 0.5 mg/kg. On the other hand the concnetrations of the residue reported in the manuscript was up to 37.38 mg/kg. There is a conflict, could please explain it or reorganised the linearity of the method.

Line 249. Since a new method has been developed, it should be discused with the other method’s performance cited in the literature.

Line 296. Could you please explain how you obtained those factors

Line 220. Since a new method has been developed, the meausurement uncertainty should be estimated.

Author Response

(The authors gave the same response as above.)
